# SARS-CoV-2 Infection in Hereditary Hemorrhagic Telangiectasia Patients Suggests Less Clinical Impact Than in the General Population

**DOI:** 10.3390/jcm10091884

**Published:** 2021-04-27

**Authors:** Sol Marcos, Virginia Albiñana, Lucia Recio-Poveda, Belisa Tarazona, María Patrocinio Verde-González, Luisa Ojeda-Fernández, Luisa-María Botella

**Affiliations:** 1Otorrinolaringology Department, Hospital Universitario Fundación Alcorcón, 28922 Madrid, Spain; solmarsal70@gmail.com; 2Molecular Biomedicine Department, Centro de Investigaciones Biológicas Margarita Salas, CSIC, 28040 Madrid, Spain; vir_albi_di@yahoo.es (V.A.); luciarecio@hotmail.com (L.R.-P.); mluisa.ojeda@gmail.com (L.O.-F.); 3CIBER Rare Diseases, U-707, Consortium CIBERER from Carlos III Institute, 28029 Madrid, Spain; 4Preventive Medicine and Public Health Department, Hospital Universitario Fundación Jiménez Díaz, 28040 Madrid, Spain; belimed22@hotmail.com; 5Health Centre Barrio del Pilar, SERMAS, 28029 Madrid, Spain; pverdegonzalez@hotmail.com; 6HHT Spanish Patients Asociation, 28040 Madrid, Spain

**Keywords:** SARS-CoV-2, COVID-19, hereditary hemorrhagic telangiectasia (HHT), pandemic, ACE2 receptor, inflammation, cytokine storm

## Abstract

At the moment of writing this communication, the health crisis derived from the COVID-19 pandemic has affected more than 120 million cases, with 40 million corresponding to Europe. In total, the number of deaths is almost 3 million, but continuously rising. Although COVID-19 is primarily a respiratory disease, SARS-CoV-2 infects also endothelial cells in the pulmonary capillaries. This affects the integrity of the endothelium and increases vascular permeability. In addition, there are serious indirect consequences, like disruption of endothelial cells’ junctions leading to micro-bleeds and uncontrolled blood clotting. The impact of COVID-19 in people with rare chronic cardiovascular diseases is unknown so far, and interesting to assess, because the virus may cause additional complications in these patients. The aim of the present work was to study the COVID-19 infection among the patients with Hereditary Hemorrhagic Telangiectasia (HHT). A retrospective study was carried out in a 138 HHT patients’ sample attending an Ear Nose and Throat (ENT) reference consult. The evaluation of the COVID-19 infection in them reveals milder symptoms; among the 25 HHT patients who were infected, only 3 cases were hospitalized, and none of them required ICU or ventilation assistance. The results are discussed in the light of macrophage immune response.

## 1. Introduction

At the time of writing this communication, the health crisis derived from the SARS-CoV-2 (COVID-19) pandemic has affected more than 120 million confirmed cases, among these, 40 million corresponding to Europe. In total, almost 3 million deaths have been registered, leading to a sanitary and economic crisis worldwide.

The figures for Spain are 3,136,321 infected and 73,000 deaths, in a total population of 47,351,567. The incidence of infection is then 6.78% of the whole population [1].

The gateway for the infection is the pharyngeal and nasal mucosa, where the virus enters through aerosol transmission or direct contact. The virus penetrates the mucosal epithelium and the endothelium through the ACE2 receptor (angiotensin convertase 2) [2]. This receptor is expressed in different cell types, notably epithelial, fibroblasts, macrophages and vascular endothelial cells.

COVID-19 is also the cause of an endothelial disease, which in the course of its progression would follow the spread in the lower respiratory tract. SARS-CoV-2 binds to the ACE2 receptor and infects endothelial cells in the pulmonary capillaries. This affects the integrity of the endothelium and increases vascular permeability. This vascular damage promotes the development of pulmonary edema and respiratory failure. Next, leukocytes (especially neutrophils) are directed towards the “activated” pulmonary endothelium. Signaling molecules, inflammatory cytokines (generated by the endothelium and the immune system cells) would increase the damage to lung tissue cells by triggering apoptotic processes [3].

In addition to direct damage to the endothelium, there are serious indirect consequences. Disruption of the junctions between endothelial cells can trigger micro-bleeds and uncontrolled blood clotting. On the other hand, blockage of small capillaries by inflammatory cells, coupled with possible thrombosis in larger vessels, can cause ischemia (decreased blood supply) in lung tissue, and even give rise to an uncontrolled inflammatory hyperactivation reaction, the “cytokine storm” [4]. Although inflammation and coagulation are essential defense mechanisms in the body, too much of them can cause irreversible and lethal damage to the patient.

The incidence of COVID-19 in cardiovascular rare diseases is unknown so far, and interesting to assess, because the virus affects the upper and lower airways but also the endothelium, leading to cardiovascular complications. The present communication deals with the incidence of COVID-19 among the patients affected by a rare vascular disease, Hereditary Hemorrhagic Telangiectasia (HHT), in a cohort of patients attending a specialized Ear Nose and Throat (ENT) consult.

Hereditary Hemorrhagic Telangiectasia (HHT) or Rendu-Osler-Weber syndrome is a genetic dominant autosomal multisystemic vascular rare disease, whose penetrance increases with age. The Curaçao criteria represent the clinical diagnosis of HHT, including its main symptoms: spontaneous and recurrent epistaxis (nose bleeds), mucocutaneous telangiectases, visceral localization (gastrointestinal telangiectases and/or arteriovenous malformations (AVMs), mainly in lung, brain or liver, and a first-degree family member with a definite diagnosis of HHT [5,6,7]. The prevalence of HHT varies between 1:5000 and 1:8000 on average, although because of the “founder effect” and “insulation effect”, the prevalence is higher in some regions such as the Jura region in France, Funen Island in Denmark and the Caribbean Dutch Antilles [7,8,9]. Heterozygous mutations in either *ENDOGLIN (ENG)* or *ACVRL1/ALK1* genes trigger the pathogenesis of HHT in over 90% of HHT patients [10,11]. Mutations in *ENDOGLIN* lead to HHT1 whereas in *ACVRL1* cause HHT2. With 93% of patients suffering light to moderate bleedings, epistaxis presents is the most frequent clinical manifestation of HHT [12,13,14]. It affects over 90% of patients before the age of 21, normally interfering with their quality of life and leading to chronic anemia. Epistaxis is due to the telangiectases of the nasal mucosa, focally dilated venules, often connected directly with dilated arterioles [15].

The aim of the present work was to study the COVID-19 infection in a cohort of 138 HHT patients attending an Ear Nose and Throat (ENT) reference doctor for HHT. HHT is a rare disease, but the ENT is the specialist visited most frequently for the epistaxis. We wanted to describe the clinical and demographic characteristics of the HHT patients affected by COVID-19, and whether there was any relation between HHT type 1 or 2 and SARS-CoV-2 infection.

## 2. Materials and Methods

### 2.1. Population under Study

The present analysis represents a retrospective observational study of a 138 cohort of HHT patients who belong to the ENT reference HHT consult, of the Health Care Provider (HCP), University Hospital, Alcorcon Foundation, (HUFA) of Madrid, Spain.

The collection of the data corresponds to the 11 first months of the pandemic (from March 2020 to February 2021). The incidence of COVID-19, among the HHT group belonging to the ENT consult, was assessed by answering a quick questionnaire, either by e-mail or by phone calls. The idea of performing such a survey came after a Webinar on HHT and COVID-19, organized by the HHT Spain patient association. Patients were previously informed about the study and signed an informed consent form. Variables collected included: sex, age, type of HHT genetic diagnosis (type 1 or 2), HHT-ESS (HHT-Epistaxis Severity Score) presence of arteriovenous malformations (AVMs), COVID-19 diagnosis, and symptoms of the SARS-CoV-2 infection, including hospitalization. Not all the patients received PCR testing, only those with symptoms or those being asymptomatic but in direct contact with relatives or friends tested positive to SARS-CoV-2. The patients diagnosed as positive of COVID-19 obtained the result by PCR; only in one case was diagnosis assessed by a serological test.

### 2.2. ELISA (Enzyme-Linked Immune Adsorbent Assay) for Detection of Inflammatory Cytokines in Macrophages of HHT Patients

The data provided for ELISA and qPCR analysis were not obtained from macrophages of patients during the COVID-19 pandemic. These data belong to RNA and culture supernatants collected before the pandemic, and belonging to an HHT sample collection of the group of research. Samples of 10 mL from peripheral blood were extracted in EDTA anticoagulant tubes after informed consent of the patients. Proteins studied were: Activin A (DAC00B), CCL20/MIP-3 α (DM3A00), IL-1β (DLB50), IL-6 (D6050), IL-12p40 (DP400), TSP-1 (DTSP10) (R&D Systems).

The detection of proteins in solution (cell culture supernatants) was performed by quantitative ELISA (Enzyme-linked immunoadsorbent assay). Commercial Quantikine^®^ Colorimetric Sandwich ELISA kits from R&D Systems (Minneapolis, MN, USA), whose reference is listed in the following table, were used. Cytokines’ production in the supernatant of cell cultures’ macrophages from HHT patients and healthy donors were analyzed after treatment with 10 ng/mL of LPS. Levels of IL-1β, IL-12p40, IL6, CCL20, TSP-1 and Activin A released into the supernatant after 48 h of culture were measured. Optical density was determined using a GloMax^®^ Multi Detection System microplate reader (Promega, Madison, WI, USA) with a 450 nm filter. The background correction wavelength was set at 540 nm, following the manufacturer’s instructions.

### 2.3. Analysis of ACE2 Expression by RT-qPCR

Cellular RNA was extracted from macrophages treated with LPS of 10 ng/mL in culture using the commercial kit NucleoSpin^®^ RNA II (Macherey-Nagel, Düren, Alemania). A total of 600 ng of total RNA was subjected to reverse transcription using the First Strand cDNA Synthesis (Roche, Mannheim, Germany), and random primers. Quantitative RT-PCR was performed from 2 microliters of cDNA, and as housekeeping gene, Actin was used. The iQTM SYBR^®^ Green Supermix (Bio-Rad, Herts, UK) was used for the quantitative PCR. ACE2 gene was amplified with the following primers designed according to the software program of the Universal Probe Library: β-actin Fwd: 5′-AGCCTCGCCTTTGCCGA-3′; β-actin Rev: 5′-CTGGTGCCTGGGGCG-3′, ACE-2 Fwd: 5′-TCCATTGGTCTTCTGTCACCCG-3′; ACE-2 Rev: 5′-AGACCATCCACCTCCACTTCTC-3′.

### 2.4. Statistical Analysis

Qualitative variables are presented with their frequency distributions. For quantitative variables, the mean and standard deviation were calculated. Quantitative variables were compared by the Student’s *t*-test; being statistically significant, those differences where *p* < 0.05. In the Figures 1 and 2, they are represented as follows: * *p* < 0.05 ** *p* < 0.01 *** *p* < 0.001. Statistical analyses were carried out with the software SPSS Windows version 11.0.0 (SPSS Inc., Chicago, IL, USA).

## 3. Results

### 3.1. SARS-CoV-2 Infection Data among a Cohort of 138 HHT Patients

The HHT patients attending the ENT consult of HUFA come from all over Spain, and they are representative of the clinical spectrum of HHT patients in Spain, concerning type of HHT, the degree of visceral involvement (AVMs) and a broad range of HHT-ESS, from severe to mild [16]. Data from the whole group of 138 patients attending the ENT reference consult are not, in general, different from the 25 HHT patients who tested positive for COVID-19, concerning the degree of HHT symptoms. As seen in Table 1, data from each single positive COVID-19 patient are shown. The range of all the considered parameters: age, sex ratio, the HHT1 vs. 2 ratio, and the presence of AVMs in these COVID-19-positive patients are not different from the whole group of patients. In particular, in Table 1, we may see 36% patients with pulmonary AVMs (PAVMs), 76% with hepatic AVMs (HAVMs), 4% cerebral AVMs. PAVMs were predominant in HHT1 (75%) vs. HHT2 (17.6%) (Table 1). These frequencies, the same as sex ratio, and prevalence of HHT2 vs. HHT1are within the range of those reported for Spain (RiHHTa Registry) including 211 patients with a mean age of 42 [17].

These patients are currently followed at the ENT consult, by sclerotherapy and propranolol cream on demand [16]. Some of them require additional treatments as mentioned in Table 1. The demographic results of the COVID-19-positive HHT population are shown in Table 2.

The frequency of positive COVID-19 HHT patients was 18.11%, with an average of age of 49 ± 18.9 years. Among the affected patients, 64% were women, and 36% men. Concerning the clinical symptoms of COVID-19-positive patients, 68% had symptoms, mostly mild/moderate, and 32% remained asymptomatic. Only 3 patients (12%) were admitted in hospital, due to pneumonia symptoms, but without needing ventilation or the intensive care unit (ICU). It is noteworthy mentioning that one of the patients was in hospital due to a serious anemia, with need of transfusions, but as a consequence of her chronic anemia due to HHT, than by COVID-19 infection. Considering in detail the symptoms shown by the 18 patients, cough was the most frequent (35.3%), followed by dyspnea (29.4%) and diarrhea (29.4%), myalgia (23.5%), headache (17.6%), fever (11.8%) and anosmia/disgeusia (11.8%). Among the COVID-19-positive patients, 68% were HHT2 and 32% were HHT1. This distribution is similar to the distribution of HHT1 versus HHT2 patients in the ENT consult, and also in agreement with the Spanish ratio with the HHT2 predominance over HHT1 [18], characteristic of Mediterranean countries. Thus, the type of HHT does not seem to affect the frequency, nor the degree of severity of the infection. All data are summarized in Table 1. In general, it does not seem to be an evident relation of HHT symptoms and COVID-19 symptoms, however, in the case of the patient with chronic anemia, COVID/SARS-CoV2 infection increased bleeding, and she required blood transfusions and hospitalization.

COVID-19 incidence in Spanish population is currently at 6.78% [19] while in our descriptive study, the COVID-19 frequency in HHT patients was 18.11%. However, these percentages are not directly comparable. The ratio should be obtained within the total HHT Spanish population, and the results are from a sample of 138 HHT. The incidence appears higher than the whole population because the rate of testing is also higher >10%. This is because the chronic condition of HHT leads to a closer clinical follow-up than in the general population. It is noteworthy mentioning that 8 out of 25 patients were asymptomatic (32%), similar to the general population asymptomatic percentage, range (21.9–35.8%). It is notorious that the percentage of hospitalizations in our study was very low, only 3 cases (12%), while in the general population, it was 40% [19]. Moreover, most importantly, only 3 cases of pneumonia were present, but without need of ventilation in the ICU. In our sample population, the most frequent symptoms were respiratory (cough and dyspnea) and digestive (diarrhea), similar to the general population. It is also noteworthy mentioning the case of one patient infected twice but asymptomatic. Regarding the treatments, people at home were only treated with paracetamol 1 g every 8 h. In the cases of pneumonia, antibiotic was added to prevent/treat concomitant bacterial infection. Altogether, the conclusion to draw is that while we cannot say that the rate of infection is different to the general population, the severity of COVID-19 seems clearly weaker.

### 3.2. ACE2 Expression in HHT and Control Macrophages

The incidence among HHT does not seem to be less than in the general population, since 25 cases were recorded in a cohort of 138 patients. Nonetheless, the expression of ACE2 receptors was studied, in HHT and control macrophages. To this purpose, real-time RT-qPCRs for ACE2 were performed in macrophages of 3 HHT1, 4 HHT2, and 4 control independent sample donors. At similar cycle for the Actin as housekeeping gene, no significant differences were found among them (Figure 1). In this figure, the higher Cycle threshold (Ct), the less is the expression of ACE2. The Ct of ACE2 ranges from 32.5 to 36 in all samples. The Ct in the case of HHT1 were more homogenous around 35.6, showing less expression than in HHT2 (*p* < 0.02), but similar to control donors. Differences among controls and HHT were not statistically significant.

### 3.3. Inflammatory Cytokines in HHT and Control Macrophages

Inflammatory response of macrophages to LPS, in HHT and control patients was analyzed as key factor involved in the so-called cytokine storm triggered by COVID-19 infection in many cases [4]. One way to assess in vitro the inflammatory response in HHT patients versus non-HHT is measuring the production of pro-inflammatory cytokines/chemokines pertinent to the cytokines observed as elevated in COVID-19 patients. For this purpose, mononuclear cells from peripheral blood were cultured from 10 HHT patients (including 5 HHT1 and 5 HHT2), and 10 control donors. The cells were cultured for 48 h in medium supplemented with LPS to induce the inflammatory response. The levels of IL-1β, IL-6, IL-12p40, and CCL20, secreted by the cells were measured by ELISA. In addition, the levels of TSP-1 (Thrombospondin), and the production of Activin A, which is of added interest, as involved in the pro-inflammatory response, were measured [20]. There were no significant differences between HHT1 and HHT2, therefore, the HHT population data were pooled versus control donors. The levels of the selected cytokines (IL-6, IL-1β, IL12p40), CCL20 and TSP-1 were decreased in HHT patients compared to the healthy donors (Figure 2). Similarly, a deficiency in Activin A production after LPS stimulation was observed in cells from HHT patients. In all cases, differences are statistically significant, with decreases of more than 50% for IL-6, IL-1β, IL12p40 and around 50% for CCL20, TSP-1 and Activin A. This significant decrease in inflammatory cytokines detected in the HHT population may explain, at least, partially, the low number of hospitalized patients (only 3 among 25), the absence of acute symptoms, and the lack of needs of ICU, and mechanical ventilation.

## 4. Discussion

There are three essential points to control the pandemic situation caused by SARS-CoV-2: (i) the early diagnosis to prevent the spread of the virus; (ii) the search for effective and safe treatments, essential to reduce the morbidity and mortality of the virus, and (iii) the development of quick vaccination plans to provide the population with immunity against the virus. Ayres (2020) [3] describes 4 phases during infection with COVID-19. Phase 1 presents a moderate symptomatology with fever, discomfort, and dry cough. Phase 2 is characterized by pneumonia with or without hypoxia. As the disease progresses, patients will develop acute respiratory syndrome with multi-organ failure and immune shock (Phase 3). Patients who recover from infection (Phase 4) show a phenotype of resistance, although some of them will never return to their normal pre-infection state (COVID persistence) [21].

The fact of having been diagnosed from HHT does not confer per se an increased risk for SARS-CoV-2 infection. On the other hand, the presented data rather suggest that the HHT patients infected by SARS-CoV-2 do not suffer a more severe infection than the general population, but rather the contrary. The degree of the COVID-19-derived infection symptoms in the HHT sample group studied seemed milder. We will try to discuss the possible factors explaining the COVID-19 less severe symptoms among HHT patients. As a previous reference, Riera et al. [22] reported only one case of a HHT2 COVID-19-positive patient admitted to hospital, in Spain, after the first wave of 2020. The patient was a woman of 74 years, and was hospitalized due to a COVID-19-derived pneumonia. Her clinical course did not involve mechanical ventilation, and she was successfully discharged after two weeks. In this letter [22], the authors refer to this case as the only HHT case admitted in hospital among the RiHHTa (Computerized Spanish Registry of Hereditary Hemorrhagic Telangiectasia). The authors hypothesize that the condition of HHT leads to a damaged endothelium with inflammation and an abnormal angiogenesis which would impair the SARS-CoV-2 infection. This fact would explain the mild clinical symptoms of COVID-19 in HHT patients.

To gain further insight into the COVID-19-HHT relationship, we have explored other factors which may have contributed to the degree of COVID-19 infection in this disease. On one hand, the ACE2 receptor expression and on the other hand, the amount of inflammatory cytokines secreted by HHT macrophages compared with control macrophages. ACE2 that is abundantly expressed in the lungs, the heart, and other tissues is used by SARS-CoV-2 as a functional receptor for their entrance into the cells [23]. The incidence among HHT does not seem to be less than in the general population. Nonetheless, the expression of ACE2 receptors in macrophages was studied, and the results revealed that there are no significant differences between control and HHT macrophages.

Another factor to take into account when examining the severity of the COVID-19 infection is the exacerbation of the inflammatory response triggered by the virus which has been defined as cytokine storm, on the immune system. The inflammatory response of control and HHT macrophages were analyzed, preferentially selecting those inflammatory cytokines reported as involved in the cytokine storm. This acute proinflammatory response damages tissues, including endothelium and contributes to the severity of the disease. The levels of the selected cytokines (IL-6, IL-1β, IL12p40), CCL20 and TSP-1 were decreased in HHT macrophages compared to the healthy donors (Figure 2). Similarly, a deficiency in Activin A production after LPS stimulation was observed in cells from HHT patients. In all cases, differences are statistically significant, with decreases of around or more than 50% in HHT samples. The pathogenesis of the acute pulmonary injury related to COVID-19 is similar to that occurring in other disorders that induce hyperinflammatory state with a release of high amounts of pro-inflammatory cytokines, mainly, IL-1, IL-6 and TNF-α. Thus, drugs that usually serve to treat rheumatic or autoimmune syndromes may play a major role in this setting [24]. In the presence of severe COVID-19 infection, targeted therapies are needed. The use of drugs with therapeutic properties already used for other therapeutic purposes, and which can therefore immediately enter clinical trials because their side effects are known, has been the strategy employed during the pandemic, or what is known as therapeutic drug repositioning [25]. With this goal of repositioning, WHO promoted the SOLIDARITY trial, which has been the largest global (30 countries) randomized drug versus control strategy. The 6-month interim analysis (15 October) indicates that remdesivir, hydroxychloroquine, lopinavir/ritonavir and interferon appear to have little or no effect on mortality or disease course [26]. Notably, Dexamethasone is the only approved drug for the treatment of severe COVID-19 patients who require oxygen therapy (from supplemental oxygen to mechanical ventilation) [27]. Therefore, there is an urgent need to continue the search for new repositioning drugs for immediate use in the absence of targeted treatment. Based on the data of this study, no further analysis regarding special treatment options in HHT could be made.

Nowadays, it is well known that COVID-19 poor prognosis is related to the most common comorbidities as hypertension, obesity, and diabetes [28]. Since in many cases, drugs which decrease cytokines, as corticoids and tocilizumab, had been used to improve the COVID-19 condition; in a way, HHT patients would be naturally producing less cytokines. Thus, without need of these treatments upon SARS-CoV-2 infection, HHT condition would avoid or smooth the acute phase, explaining the milder infections suffered by them.

In this sense, Figure 3 represents a graphical hypothesis of immune response in HHT and control population highlighting the decreased cytokine storm triggered in HHT patients after COVID-19 infection. We believe this may be, among other factors, a crucial point to explain the milder symptoms detected in HHT. In a similar way, we could hypothesize that patients with autoimmune diseases under anti-inflammatory treatment might be, to a certain extent, protected from the severe phases of COVID-19. This hypothesis should be analyzed when data from autoimmune disease cohorts and the COVID-19 infections will be published. On the other hand, in a mouse model KO for endoglin in the myeloid linage, KO mice were protected compared to their wild type and heterozygote littermates, following an in vivo septic shock by LPS. In fact, the survival was higher, and the first deaths were delayed by 36 h compared to their wild type littermates’ linage [29].

Another disease where the HHT condition may confer a better outcome is cancer. In a mouse model of skin carcinoma, HHT mice developed fewer tumors then their wild type littermates. These studies suggest that endoglin behaved as a suppressor of malignancy in experimental and human epithelial carcinogenesis, although it could also promote metastasis in other types of cancer [30]. In humans, it has been hypothesized that individuals with HHT may be protected against life-limiting cancers [31] due to limited angiogenesis, since endoglin and ALK1 are proangiogenic factors. Anti-endoglin and anti-Alk1 therapies have been used for targeting tumors [32].

As a summary, the results of this communication suggest that HHT patients might be protected from developing a cytokine storm after SARS-CoV-2 infection. The results of this work represent, up to the moment, the largest series of HHT patients diagnosed of/affected by COVID-19 published to date. We are planning to conduct a larger survey with data derived from the HHT patient associations, at national and international levels, to gain more insights into HHT and the COVID-19 infection, and possibly reinforce this hypothesis.

### Limitations and Interest of the Study

The main limitation of our study is that the sample analyzed represents a retrospective study of a group of patients belonging to a reference HHT ENT consult who attend regularly to this HCP, and not a random sample. Therefore, the results may not be representative for the whole HHT population, and the whole pandemic situation, but provide interesting interim results. The reason for focusing the survey on the patients attending the ENT consult is due to the fact that epistaxis is the most prevalent symptom in HHT, and moreover, this ENT consult is taking care of HHT patients from all over Spain, due to the degree of satisfaction of the HHT patients. Maybe the incidence of COVID-19 among these patients was even higher since not all received PCR testing. Asymptomatic COVID-19-positive patients could have been categorized as false negative.

We must realize that in rare diseases, the low prevalence of patients make this type of study difficult. We are aware that this is only a starting point to draw conclusions which may be confirmed by larger studies. On the other hand, this study presents the novelty of being a study conducted in an emergency situation. Perhaps it represents the first study where SARS-CoV-2 infection has been investigated in a cohort of patients with a rare disease.

## Figures and Tables

**Figure 1 jcm-10-01884-f001:**
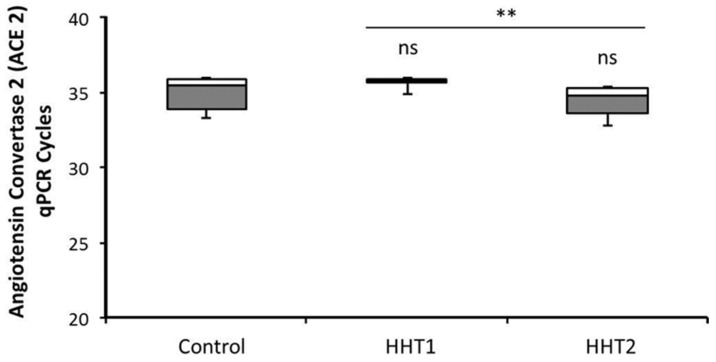
RT-qPCR of Angiotensin Convertase Enzyme type 2 in macrophages of HHT1, HHT2 and Control donors. Mononuclear cells from healthy and HHT donors were cultivated in DMEM with 10% FCS, and treated by LPS 10 ng/mL for 48 h. Cells were lysed, and RNA was extracted as described in Materials and Methods. RT-qPCR was performed for ACE2 receptor in 4 HHT2, 3 HHT1 and 4 control donor samples. Figure represents the number of cycles corresponding to the Ct of the amplification curve at similar Actin RNA amount taken as housekeeping gene. Ns—not significant; ** *p* < 0.01.

**Figure 2 jcm-10-01884-f002:**
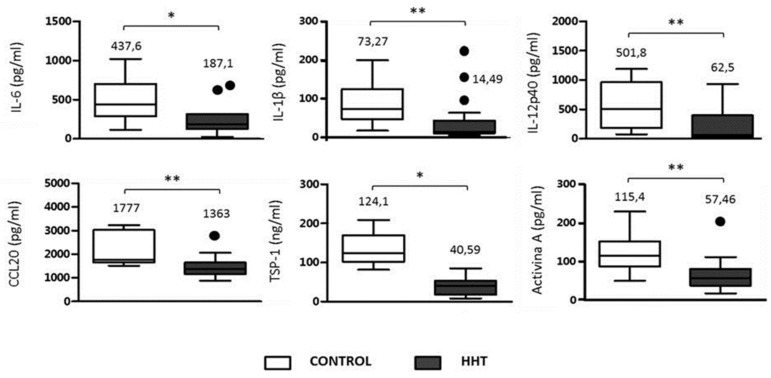
Inflammatory cytokines’ analysis in culture media of macrophages, isolated from HHT and control donors. A total of 500,000 mononuclear cells from healthy and HHT donors were cultivated in DMEM with 10% of FCS, and treated by 10 ng/mL of LPS for 48 h. The production of cytokines/chemokines was measured by the corresponding ELISA kits, as described in Materials and Methods. * *p* < 0.05; ** *p* < 0.01.

**Figure 3 jcm-10-01884-f003:**
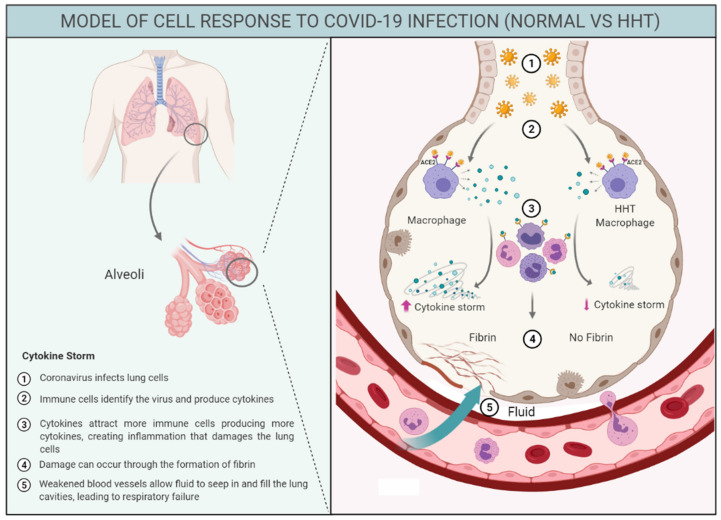
Graphical scheme explaining the COVID-19 reaction in HHT macrophages compared to macrophages of general population. The figure is based on the hypothesis that the cytokine production in HHT is not as exacerbated as the frequently occurred in the normal population leading to the “so-called” cytokine storm. 1. Coronavirus infects lung cells; 2. Immune cells, includes macrophages, identify the virus and produce cytokines; 3. Cytokines attract more immune cells, such as white blood cells, which in turn produce more cytokines, creating a cycle of inflammation that damages the lung cells; 4. Damage can occur through the formation of fibrin; 5. Weakened blood vessels allow fluid to seep in and fill the lung cavities, leading to respiratory failure.

**Table 1 jcm-10-01884-t001:** HHT characteristics and COVID-19 symptoms in the affected group. Results related to HHT symptoms in the positive COVID-19 cases, and the COVID-19 infection derived symptoms.

Patient	Gender	Gene	Type	AVMS	HHT-ESS	Age	Symptoms	Hospital
#1	♀	ENG	HHT1	HAVM	0.91	37	asymptomatic	-
#2	♂	ENG	HHT1	PAVM, HAVM	1.41 ^α^	74	asymptomatic	-
#3	♀	ALK1	HHT2	HAVM	1.01	57	headache, diarrhea	-
#4	♂	ALK1	HHT2	HAVM	1.91	64	headache, diarrhea, myalgias	-
#5	♂	ENG	HHT1	PAVM, HAVM	0.51	70	cough, shortness of breath	-
#6	♂	ALK1	HHT2	HAVM	7.46 ^ε^	63	suspected diarrhea, serologic detection months later	-
#7	♀	ALK1	HHT2	HAVM	3.51	63	myalgias, headache, nosebleed, diarrhea	-
#8	♂	ALK1	HHT2	-	0.0	29	asymptomatic	-
#9	♀	ENG	HHT1	PAVM, CAVM, HAVM	1.41	32	asymptomatic	-
#10	♀	ALK1	HHT2	PAVM, HAVM	2.43	47	pneumonia, cough, dyspnea	YES
#11	♀	ALK1	HHT2	HAVM	1.41	56	asymptomatic	-
#12	♀	ENG	HHT1	PAVM, HAVM	1.41	62	asymptomatic	-
#13	♀	ALK1	HHT2	HAVM	3.33	70	asymptomatic	-
#14	♀	ALK1	HHT2	HAVM	5.18 ^γ^	87	pneumonia, anemia	YES
#15	♂	ALK1	HHT2	HAVM	0.0	55	headache, diarrhea	-
#16	♀	ALK1	HHT2	-	0.0	25	anosmia, headache	-
#17	♀	ALK1	HHT2	PAVM, HAVM	3.33	50	anosmia, headache, diarrhea	-
#18	♀	ENG	HHT1	PAVM, HAVM	3.33	52	pneumonia, no dyspnea	YES
#19	♀	ENG	HHT1	PAVM, SpAVM	0.0	16	asymptomatic	-
#20	♀	ENG	HHT1	HAVM	2.43 ^Δ^	49	anosmia, ageusia, moderate fever and slight muscular pain, diarrhea	-
#21	♂	ALK1	HHT2	-	0.0	41	anosmia, ageusia, moderate fever and slight muscular pain	-
#22	♂	ALK1	HHT2	-	0.0	18	infected twice, rhinitis	-
#23	♀	ALK1	HHT2	HAVM	0.51	50	tonsil and ear infections, cough, fever, pain in the chest, and low oxygen saturation (91%)	-
#24	♀	ALK1	HHT2	-	0.0	16	fever, vomit, tiredness and breathless	-
#25	♂	ALK1	HHT2	HAVM	0.0	63	anosmia, headache	-

^α^ #2 at the beginning, need transfusions; in the last year, no need of transfusions. ^ε^ #6 dependent on transfusions before Young’s procedure. Reopening of Young’s in the last year, bleeding but no need of transfusions yet. ^γ^ #14 chronic anemia, needed transfusion upon hospitalization. ^Δ^ #20, treated with low dose of tacrolimus and sclerotherapy. Before, she was transfusion-dependent.

**Table 2 jcm-10-01884-t002:** Characteristics of patients with HHT and COVID-19. Results in percentages of the different demographic characteristics of the affected population. * Figures represent mean (±SD).

	HHT1 *n* (%)	HHT2 *n* (%)
Sex		
Male Female	2 (25) 6 (75)	7 (41.1) 10 (58.8)
Age	49 (±19.9) *	49.3 (±19.6) *
Symptoms		
No	5 (62.5)	3 (18.8)
Yes	3 (37.5)	14 (81.2)
Hospitalization		
No	7 (87.5)	15 (87.5)
Yes	1 (12.5)	2 (12.5)

## Data Availability

Reported results can be found in the files of Centro de Investigaciones Biológicas Margarita Salas (CIB, CSIC) and in the files of Hospital Universitario Fundación Alcorcón (HUFA), Madrid, Spain.

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
