# Peer review of "SARS-CoV-2 Infection in Hereditary Hemorrhagic Telangiectasia Patients Suggests Less Clinical Impact Than in the General Population"

_jcm, 2021, doi:10.3390/jcm10091884_

Round 1

Reviewer 1 Report

The authors present a novel and very interesting theory however there are concerns regarding the methods and conclusions: 

Introduction:

  1. Lack references:
    1. Reference to the Spanish numbers. Date it is related to.
    2. Lines 53-77 require several references
  2. Introduction is too long, part of it is irrelevant to the study and part should be moved to the discussion

Methods:

  1. Study population- it is not clear what was the study group- ambulatory patients who had a scheduled visit? Urgent visits due to epistaxis? Other? When was the PCR done? In clinic while visiting? These details are crucial to understand the characteristics of the study group. If they were diagnosed while in clinic there is a significant selection bias because only presumed healthy patients came. In any case it seems that this is not a representative group- not of the general population and not of the HHT population. Patients attending the ENT clinic are suffering moderate to severe epistaxis while it is well known that a majority of HHT patients are having mild-moderate epistaxis. In addition:
    1. Patients attending the ENT clinic are probably not febrile (I assume febrile patients could not enter the clinic) and not suffering other symptoms. Sick patients (with COVID19) stayed home.
    2. If a scheduled HHT patient is hospitalized with COVID19 - he probably cancelled his visit and did not show up.
    3. HHT patients attending the clinic are older than the general and the HHT population as children and young adults suffer less epistaxis
  2. The methods should describe the study population- is it :"all patients attending the ENT clinic in February 2021" or is it only part of the patients? Whom? The actual numbers should be in the results

Results:

  1. The results should contain the main results- how many patients were included? How many were diagnosed with COVID. The numbers (138;25) should be in the results section and not in the methods.
  2. Data regarding the manifestations of the study population should be presented with numbers preferably in a table: ESS- mean/median; range; SD. AVMs prevalence etc.
  3. The authors mention that several clinical parameters are similar to the general Spanish HHT population "they are representative of the clinical spectrum of HHT patients in Spain", however no numbers are presented (not of the study group and not of the general HHT population) and no P value is calculated. It should be noted that in the limitations the authors clearly say that this is not a representative HHT population
  4. If the purpose of table 1 is to demonstrate that there is no difference between HHT1 and 2 a p value should be calculated.
  5. Data regarding COVD19 in the general Spanish population require references- 32% asymptomatic. 40% hospitalization?

Discussion:

  1. The authors can not say anything about the prevalence of COVID in HHT (in contrary to the 1st sentence in the discussion). Probably we can not say also anything about having a milder course because of the selection bias
  2. The second part of the study looking for ACE2 and proinflammatory cytokines should be in the results section as the authors are reporting an in vitro study. The interpretation should be in the discussion.
  3. The authors suggest a very interesting hypothesis- however – inflammatory responses, cytokines storms and other IL6 and other cytokines responses are not unique to COVID19 – is there any evidence from other diseases that the HHT response is different from the general population- if so- please reference and if not please provide an explanation to the reason why it is only in COVID19?
  4. Fig 2 – what does * and ** mean? What is the p value?

Limitations- should be at the end of the discussion.

Author Response

Dear reviewer,

I feel really grateful for your comments which helped a lot in improving the manuscript, and clarifying several points that were not clear in our previous version.

Thank you very much. We hope the manuscript looks now much better

Please, find here, point by point the answers to your questions:

Comments and Suggestions for Authors

The authors present a novel and very interesting theory however there are concerns regarding the methods and conclusions:

Introduction:

  Lack references:

 Reference to the Spanish numbers. Date it is related to

Thank you very much for this observation. It is really necessary to include the source of the data from the Spanish Health Department, including also the date. We have included this in  the Introduction, and in the Results section (References 1 and 20, respectively)

Lines 53-77 require several references

Thank you very much, references 2 and 4 were added corresponding to the coronavirus infection, ACE2 receptor, and as the involvement of vasculature.

 Introduction is too long,  part of it is irrelevant to the study and part should be moved to the discussion OK part of it was moved to discussion, and part eliminated

Thank you. Accordingly, part of the introduction was moved to discussion, and it was shortened eliminating irrelevant parts.

Methods:

Study population- it is not clear what was the study group- ambulatory patients who had a scheduled visit? Urgent visits due to epistaxis? Other? When was the PCR done? In clinic while visiting? These details are crucial to understand the characteristics of the study group. If they were diagnosed while in clinic there is a significant selection bias because only presumed healthy patients came. In any case it seems that this is not a representative group- not of the general population and not of the HHT population. Patients attending the ENT clinic are suffering moderate to severe epistaxis while it is well known that a majority of HHT patients are having mild-moderate epistaxis. In addition:

    Patients attending the ENT clinic are probably not febrile (I assume febrile patients could not enter the clinic) and not suffering other symptoms. Sick patients (with COVID19) stayed home.

    If a scheduled HHT patient is hospitalized with COVID19 - he probably cancelled his visit and did not show up.

  HHT patients attending the clinic are older than the general and the HHT population as children and young adults suffer less epistaxis

The methods should describe the study population- is it :"all patients attending the ENT clinic in February 2021" or is it only part of the patients? Whom? The actual numbers should be in the results

Thank you very much for this point. It seems that methods were not appropriately described and the actual methodology of the study was not understood. To  clarify this part, we have rewritten the section corresponding to the population used in the study, and how the data were obtained, in addition Table 1 has been added showing the COVID positive sample.

The HHT patients attending the ENT consult of HUFA, come from all over Spain, and they are representative of the clinical spectrum of HHT patients in Spain, concerning type of HHT, the degree of visceral involvement (AVMs) and a broad range of ESS-HHT (epistaxis severity score for HHT), from severe to mild [16]. Data from the whole group of 138 patients attending the ENT reference consult are not in general different from the 25 HHT patients who tested positive for COVID, concerning the degree of HHT symptoms. As seen now in Table 1, data from each single positive COVID patient are shown. The range of all the considered parameters: age, sex ratio, the HHT1vs 2 ratio, and the presence of AVMs in these COVID positive patients is not different from the whole group of patients

These patients are currently followed at the ENT consult, by sclerotherapy and propranolol cream on demand. Some of them, require additional treatments as mentioned in Table 1. Table 1 collects the results related to HHT symptoms in the positive COVID cases, and the COVID-19 infection derived symptoms. Table 2 is summarizing  in percentages, the different demographic characteristics of the affected population.

Results:

The results should contain the main results- how many patients were included? How many were diagnosed with COVID. The numbers (138;25) should be in the results section and not in the methods.

Data regarding the manifestations of the study population should be presented with numbers preferably in a table: ESS- mean/median; range; SD. AVMs prevalence etc.

    The authors mention that several clinical parameters are similar to the general Spanish HHT population "they are representative of the clinical spectrum of HHT patients in Spain", however no numbers are presented (not of the study group and not of the general HHT population) and no P value is calculated. It should be noted that in the limitations the authors clearly say that this is not a representative HHT population

    If the purpose of table 1 is to demonstrate that there is no difference between HHT1 and 2 a p value should be calculated.

We are especially grateful for the comments of the reviewer in this part, since we have reorganized the paper, and now it is more complete, and easier to understand. The results section has been completely modified, now, all the patients included and diagnosed appear in results. A new Table, Table 1 describing the manifestations of the ESS, and AVMs, and COVID symptoms has been included for the COVID positive patients. The reference numbers of ESS for the group of patients of this consult is given and referred to a published paper, ref. [18]. And we hope that all the concerns raised by the reviewer, have been clarified, and addressed in the new results section.

Data regarding COVD-19 in the general Spanish population require references- 32% asymptomatic. 40% hospitalisation?

Reference 18 was added.

Discussion:

    The authors can not say anything about the prevalence of COVID in HHT (in contrary to the 1st sentence in the discussion). Probably we can not say also anything about having a milder course because of the selection bias

    The second part of the study looking for ACE2 and proinflammatory cytokines should be in the results section as the authors are reporting an in vitro study. The interpretation should be in the discussion.

Again we thank the reviewer by the wise comments. We have avoided saying anything about the prevalence of COVID. Only we say that: “The degree of the COVID derived infection symptoms in the HHT sample group seems milder”

    The authors suggest a very interesting hypothesis- however – inflammatory responses, cytokines storms and other IL6 and other cytokines responses are not unique to COVID19 – is there any evidence from other diseases that the HHT response is different from the general population- if so- please reference and if not please provide an explanation to the reason why it is only in COVID19?

This is a very interesting point, and we have tried to address it looking in the literature. In order to discuss this point we have added the following paragraph, with the corresponding references:

In a similar way, we could hypothesize that patients with autoimmune diseases under anti-inflammatory treatment might be, to a certain extent, protected from the severe phases of COVID infection. This hypothesis should be analyzed when data from autoimmune diseases cohorts and the Covid-19 infections will be published. On the other hand in a mouse model KO for endoglin in the myeloid linage, KO mice were protected compared to their wild type and heterozygote littermates, following an in vivo septic shock by LPS. In fact, the survival was higher, and the first deaths were delayed by 36 hours compared to their wild type littermates linage [28].

Other disease where the HHT condition may confer a better outcome is cancer. In a mouse model of skin carcinoma, HHT mice developed less tumors that their wild type littermates. These studies suggest that endoglin behaved as a suppressor of malignancy in experimental and human epithelial carcinogenesis, although it could also promote metastasis in other types of cancer [29] In humans, it has been hypothesised that individuals with HHT may be protected against life-limiting cancers [30] due to limited angiogenesis, since Endoglin and ALK1 are proangiogenic factors, and anti-endoglin, anti-Alk1 therapies have been used for targeting tumors [31].

Fig 2 – what does * and ** mean? What is the p value?

Thank you for the remark, we have added in the figure legend:

*p<0.05; **p<0.01

Limitations- should be at the end of the discussion.

Thank you for the remark, we have moved limitations to the end of the discussion.

Reviewer 2 Report

Using a questionnaire the authors analyzed descriptive data of patients with HHT about COVID. 25 out of 138 patients with HHT suffered from COVID and in most cases only mild symptoms of COVID were reported. About one third (8/25) did not show any symptoms and the hospitalization rate was very low, too. In addition, a quantitative ELISA of different proteins and analysis of macrophages in patients with HHT compared to controls was presented in this study. As the authors stated, there has been very few studies analyzing COVID in patients with rare diseases like HHT. Therefore, these findings are very important and the study presents interesting results; I enjoyed reading it!  However, I have a few questions/ remarks the authors could please comment on: 

  1. I would recommend to shorten the introduction especially lines 35-93.
  2. Did all 138 patients with HHT receive a PCR testing? Thus 113 patients had a negative PCR test? Did these patients have any COVID symptoms at all?
  3. Please state the timepoint at which the blood taken and analyzed in this study - During an active COVID infection/ within the HHT Group at the same time points (e.g. x days after first symptoms or directly after hospitalization...)? If the blood samples were taken at different time points this might influence the data the authors presented and should be discussed in the limitation section. 
  4. Do the authors have some data (e.g. age, sex, genetic mutation) about the whole study cohort? (like it is presented in table 1 for only 25 patients with HHT and COVID). Especially the average age of the whole study population could be interesting. Does the age significantly differ from the age of the 25 COVID+- patients with HHT? 
  5. Could the authors please state if in the general population patients with COVID at the age around 50 years suffer more often from mild or severe symptoms? Maybe the age of the presented study population also influences their COVID symptoms. This should be taken into consideration.
  6. Do the authors have any data on how HHT symptoms/ manifestations (organ manifestation, hemoglobin level/ anemia, ESS) may influence COVID-symptoms? 
  7. I guess mean +/- standard deviation is shown in table 1 ("Age")? Could the authors please add this information to the table/legend?
  8. In Figure 1 and 2 the authors compare patients with HHT with a control group. Could the authors please state if the control group was of similar age and sex? Who was included in this control group: non-HHT-non-COVID or non-HHT-but COVID+? An additional comparison of theses groups in this study or from the literature would be interesting if possible. 
  9. Could the authors please add the number of persons analyzed in figure 2 (e.g. in the legend)?  Please add description of the box plots to the legend. 
  10. Do the authors have any data about the treatment of COVID? As this may influence the presented data. Could the authors please comment on this?
  11. I would recommend to rewrite the discussion as the authors wrote some of the results in this section. To my point of view, Figure 1 and 2 belong to the result section. 
  12. Usually the study limitations are found in the discussion part. In addition, I would recommend to discuss another bias: In this study, patients with HHT were attending the ENT department/ consultant. Maybe patients with more severe symptoms directly consulted their GP/ hospital and might therefore not be included in this study. What was the reason the patients analyzed in this study consulted their ENT? COVID symptoms or other health problems?
  13. line 59: COVID-19 is also the cause (please add: "of") an endothelial....
  14. lines 166-167: what was the mean ESS?
  15. lines 188- 194: 2x "noteworthy" and 1x "of note"--> maybe the authors could improve this part?
  16. lines 195-197: I would recommend revising this sentence
  17. line 209: "diagnosed with" (instead of "from")?
  18. line 211: "suffer (please add "from") a more .... "
  19. Overall, it might be helpful if a native English speaker could also edit the manuscript. 

Author Response

Dear reviewer, we thank you by your comments and suggestions which helped a lot to improve the manuscript. We hope the present form of the manuscript will be acceptable for your, and you willl enjoy its reading. Thanks a lot.

Principio del formulario

Using a questionnaire the authors analyzed descriptive data of patients with HHT about COVID. 25 out of 138 patients with HHT suffered from COVID and in most cases only mild symptoms of COVID were reported. About one third (8/25) did not show any symptoms and the hospitalization rate was very low, too. In addition, a quantitative ELISA of different proteins and analysis of macrophages in patients with HHT compared to controls was presented in this study. As the authors stated, there has been very few studies analyzing COVID in patients with rare diseases like HHT. Therefore, these findings are very important and the study presents interesting results; I enjoyed reading it!  However, I have a few questions/ remarks the authors could please comment on: 

  1. I would recommend to shorten the introduction especially lines 35-93

Ok, this has been done

  1. Did all 138 patients with HHT receive a PCR testing? Thus 113 patients had a negative PCR test? Did these patients have any COVID symptoms at all?

This has been clarified in the Mat and Methods

  1. Please state the timepoint at which the blood taken and analyzed in this study - During an active COVID infection/ within the HHT Group at the same time points (e.g. x days after first symptoms or directly after hospitalization...)? If the blood samples were taken at different time points this might influence the data the authors presented and should be discussed in the limitation section.

The data provided for ELISA and qPCR analysis, were not obtained from macrophages of patients during the pandemic COVID. These data belong to RNA and culture supernantants collected before the pandemia, and belonging to an HHT sample collection belonging to the group of research.

  1. Do the authors have some data (e.g. age, sex, genetic mutation) about the whole study cohort? (like it is presented in table 1 for only 25 patients with HHT and COVID). Especially the average age of the whole study population could be interesting. Does the age significantly differ from the age of the 25 COVID+- patients with HHT? 

Data from the whole group of 138 patients are not significantly different from the 25 HHT patients who tested as COVID positive. As seen now in Table 1, data from each single positive COVID patient is shown. The range of all the considered parameters: age, sex ratio, the HHT1/2 ratio, and the presence of AVMs in these COVID positive patients is not different from the whole group of patients attending the ENT reference consult.

  1. Could the authors please state if in the general population patients with COVID at the age around 50 years suffer more often from mild or severe symptoms? Maybe the age of the presented study population also influences their COVID symptoms. This should be taken into consideration.

In our case the average age of HHT COVID positive patients was of 50, but with a broad range covering from teenagers, to patients older than 80 years, therefore, we believe the COVID symptoms are not influenced by age in our sample, since the Standard Deviation is almost 20, and as seen in table 1 affected COVID patients are from 16 to 87 years old.

  1. Do the authors have any data on how HHT symptoms/ manifestations (organ manifestation, hemoglobin level/ anemia, ESS) may influence COVID-symptoms?

 It does not seem to be an evident relation of those HHT symptoms and COVID symptoms, however, in the case of a patient with chronic anemia, after COVID bleeding increased and she required blood transfusions and hospitalization.

  1. I guess mean +/- standard deviation is shown in table 1 ("Age")? Could the authors please add this information to the table/legend?

Yes, thank you very much for the remark, we have added this information

  1. In Figure 1 and 2 the authors compare patients with HHT with a control group. Could the authors please state if the control group was of similar age and sex? Who was included in this control group: non-HHT-non-COVID or non-HHT-but COVID+? An additional comparison of theses groups in this study or from the literature would be interesting if possible. 

As the controls and the HHT samples were collected before the pandemia, all samples are non-COVID. The range of age and sex is similar in controls and HHT patients.

  1. Could the authors please add the number of persons analyzed in figure 2 (e.g. in the legend)?  Please add description of the box plots to the legend. 

OK this is done

  1. Do the authors have any data about the treatment of COVID? As this may influence the presented data. Could the authors please comment on this?

No special treatment, other than paracetamol, and antibiotic in the cases of pneumonia was used for COVID. In the case of patient #14 transfusion was needed

  1. I would recommend to rewrite the discussion as the authors wrote some of the results in this section. To my point of view, Figure 1 and 2 belong to the result section

Thank you very much for this suggestion. It  has been done

  1. Usually the study limitations are found in the discussion part. In addition, I would recommend to discuss another bias: In this study, patients with HHT were attending the ENT department/ consultant. Maybe patients with more severe symptoms directly consulted their GP/ hospital and might therefore not be included in this study. What was the reason the patients analyzed in this study consulted their ENT? COVID symptoms or other health problems?

OK, limitations were added at the end of the discussion. Data from patients belonging to an ENT consult were used since  epistaxis is the most generally consulted specialist in Spain, and this consult is not only local, but is taking care of patients from all over Spain.  We have explained this fact in the limitations.

  1. line 59: COVID-19 is also the cause (please add: "of") an endothelial....
  2. lines 166-167: what was the mean ESS?
  3. lines 188- 194: 2x "noteworthy" and 1x "of note"--> maybe the authors could improve this part?
  4. lines 195-197: I would recommend revising this sentence
  5. line 209: "diagnosed with" (instead of "from")?
  6. line 211: "suffer (please add "from") a more .... "
  7. Overall, it might be helpful if a native English speaker could also edit the manuscript. 

Thank you very much, we have tried to do all the suggested changes, and the manuscript was reviewed by a native  English speaker

Round 2

Reviewer 1 Report

A very interesting and novel study- however the conclusions can not be drown from the data presented in the study.

The title –"SARS-Cov-2 Infection in Hereditary Hemorrhagic Telangiectasia patients seems milder than in the general population"- I don't think the authors can conclude it is milder from 25 patients with multiple bias.

On top of this - the authors try to compare hospitalization rate, death rate between the study group and the general population- however the study group is characterized by a certain age group 49+-19y. It can not be compared to the general population where majority of deaths were in patients 80 years and older. To prove their hypothesis they should look at the same age group in the general population.

Line 25- The incidence of infection is then a 6.78% of the whole population

Methods:

-Are there only 138 HHT patients followed at the ENT clinic from the whole country? There is an estimation of 8000 HHT patients in Spain.

 -How many patients were approached? What is the response rate? Are there many patients who did not respond? Do we know what is their status? is there a bias?

Results:

Of the 138- how many were tested for covid19? Is the positivity rate similar to the general population? WHO suggested a rate of less than 10% as a marker of appropriate number of tests. It seems that in this group rate is much higher (25/less than 138).

-line 179- "COVID-19 prevalence in Spanish population is currently of 6.78% [18] while in our 179 descriptive study, the COVID-19 frequency in HHT patients was of 18.11%. However, these percentages are not directly comparable. The ratio should be obtained within the total HHT Spanish population, and the results are from a sample of 138 HHT"- the authors have to decide whether this population is representative or biased…. Results suggest that covid 19 is more prevalent in HHT than in the general population.

Line 137-"Data from the whole group of 138 patients attending the ENT reference consult are not in general different from the 25 HHT patients who tested positive for COVID-19, concerning the degree of HHT symptoms"- where is the data? Any statistics to prove it?

-line 144-"age, sex ratio, the HHT1vs 2 ratio, and the presence of AVMs in these COVID-19 positive patients are not different from the whole group of patients"- where is the data? Any statistics to prove it?

Author Response

very interesting and novel study- however the conclusions can not be drown from the data presented in the study.

Thank you very much for your kind words

The title –"SARS-Cov-2 Infection in Hereditary Hemorrhagic Telangiectasia patients seems milder than in the general population"- I don't think the authors can conclude it is milder from 25 patients with multiple bias.

Thank you, you are right, but we are stating that in 25 patients HHT and COVID positive, the symptoms of COVID were milder than those observed in the general population. Twenty 25 may seem few patients, but enough to be sure that the percentage of hospitalization only 3 in 25 is a 12% well below the 40% of hospitalization registered in Spain. None of them needed ICU or ventilation, and this is really clearly different from the general population.  We have changed slightly the title our suggestion could be:

 “SARS-CoV-2 Infection in Hereditary Hemorrhagic Telangiectasia patients suggests less clinical impact than in the general population”.

On top of this - the authors try to compare hospitalization rate, death rate between the study group and the general population- however the study group is characterized by a certain age group 49+-19y. It can not be compared to the general population where majority of deaths were in patients 80 years and older. To prove their hypothesis they should look at the same age group in the general population.

Thank you very much, this is our answer

In the group of HHT-COVID positive patients, we have the whole range from teenagers to really old, having all the ages covered by chance (they were the infected positive patients): 2 teenager (16-19), 2 in (25-29), 2 in thirties, 3 in forties, 5 in fifties, 5 sixties, 3 of seventies, 3 in the eighties. More than half were older than 50, more favorable to suffer from more severe symptoms, only due to the age, as in the general population.

In general the data related to the age would be even more prone to suffer from moderate to severe symptoms 11 older than 60.

Line 25- The incidence of infection is then a 6.78% of the whole population

OK thank you, the change has been done.

Methods:

-Are there only 138 HHT patients followed at the ENT clinic from the whole country? There is an estimation of 8000 HHT patients in Spain.

The estimation based on statistics is around 8.000, out of this, in Sierrallana/Valdecilla (Cantabria), Dr Roberto Zarrabeitia has a database of around 580 HHT positive diagnosed patients, from all over Spain. In general, there is the RiHHTa Spanish HHT registry with 211 patients registered new reference included (ref 17). The ENT consult with 138 patients from different places of Spain, is the single speciality consult accumulating more HHT patients in Spain.  This is an amazing accumulation of patients in a single consult for a rare disease, and of course the common link is epistaxis, but we think is a good group to have a first estimation of the impact (measured as severity of COVID-19 symptoms) in HHT.  In the Spanish HHT association we have 250 patients, and we have planned a survey with the same objective, see the impact of COVID-19 among HHT patients of the HHT Spanish association.

 -How many patients were approached? What is the response rate? Are there many patients who did not respond? Do we know what is their status? is there a bias?

All the 138 patients, the total group of them were approached, and the response was obtained in all cases. Of course, this was not at the first attempt, in many cases repeated phone calls or e-mails were requested. The current status of all of them is satisfactory, no COVID persistence, at least until one month ago.

Results:

Of the 138- how many were tested for covid19? Is the positivity rate similar to the general population? WHO suggested a rate of less than 10% as a marker of appropriate number of tests. It seems that in this group rate is much higher (25/less than 138).

For sure 25 were tested for COVID. It is very likely that other among the group o 138 were also tested with negative result. We can think of bias: those which could be asymptomatic and positive for COVID-19 but who were never tested by the absence of symptoms (therefore false negative). In this case, this would increase the number of asymptomatic and supporting the milder infection. 25 out of 138 is a rate of 18.1 % of PCR then over the rate of 10% considered as appropriate number of tests

-line 179- "COVID-19 prevalence in Spanish population is currently of 6.78% [18] while in our 179 descriptive study, the COVID-19 frequency in HHT patients was of 18.11%. However, these percentages are not directly comparable. The ratio should be obtained within the total HHT Spanish population, and the results are from a sample of 138 HHT"- the authors have to decide whether this population is representative or biased…. Results suggest that covid 19 is more prevalent in HHT than in the general population.

This is what we mention, that it seems that the incidence of COVID is higher in HHT, but here there is a clear bias. This is because the chronic HHT condition leads to a closer clinical contact follow up and tests than in the whole population. We have included this reason in Results line 138:

“The incidence appears higher than the whole population but because the rate of testing is also higher > 10%. This is because the chronic condition of HHT leads to a closer clinical follow-up than in the general population”.

-Line 137-"Data from the whole group of 138 patients attending the ENT reference consult are not in general different from the 25 HHT patients who tested positive for COVID-19, concerning the degree of HHT symptoms"- where is the data? Any statistics to prove it?

-Line 144-"age, sex ratio, the HHT1vs 2 ratio, and the presence of AVMs in these COVID-19 positive patients are not different from the whole group of patients"- where is the data? Any statistics to proveit?

Answering this question and the previous one (Line 137-“Data from…”) we have included a new paragraph, and new reference concerning the general HHT Spanish population in results. Section 3.1, to address both points of line 137 and 144.

“In particular in table 1 we may see, 36% patients with pulmonary AVMs (PAVMs), 76% with hepatic AVMs (HAVMs), 4% Cerebral AVMs. PAVMs were predominant in HHT1 (75%) vs HHT2 (17.6%). These frequencies, the same as sex ratio, and prevalence of HHT2 vs HHT1 is within the range of those reported for Spain (RiHHTa Registry) including 211 patients with a mean age of 42 [17]”.

Reviewer 2 Report

I thank the reviewer for implementing the requested changes and think that the manuscript has improved a lot. However, a few comments/ remarks are left: 

  1. Thank you for clarifying the study methods. As not all patients received PCR testing maybe asymptomatic COVID positive patients were categorized as false negative. This should be discussed in the study limitations.
  2. Table 2: Thank you for adding the legend. The numbers seem to be moved and not at the same level, maybe the authors can improve the layout.
  3. Line 142: Delete „now“
  4. Lines 148-150: This is rather the text for the legends of the tables 1 and 2 than main-text. I would recommend to put these sentences in the legends and write the main results of table 1 and 2 in the text with referring to the tables.
  5. Line 160 and lines 196 ff.: As mentioned above the authors might have analyzed maybe false negative results as these patients might have had COVID but were asymptomatic. Therefore, the prevalence of COVID positive HHT patients might be even higher. It would be helpful to write the number of COVID- tests that were done and how many were positive.
  6. Line 196: Here, the authors write about the incidence. Before and after they state the prevalence of COVID and HHT. Please clarify which is the correct term for the analysis the authors did.
  7. Line 257: Please change to “seemed”
  8. Line 281: Please write “were analyzed”
  9. Lines 291 ff: It is an interesting part of the discussion but the authors only write a general statement on treatments options for COVID-infections. I would recommend to either discuss something also related to HHT and the presented data or at least write that based on the data of this study no further analysis regarding special treatment options in HHT could be made.
  10. Line 305: Please add “Nowadays, it is well known..”
  11. Line 306: Please correct “common” and delete “were the most common comorbidities”
  12. Lines 307 ff.: Please revise this very long sentence. Maybe put the information into more but shorter sentences and maybe correct the placement you used the comma.
  13. Line 356: Delete „that“

Author Response

I thank the reviewer for implementing the requested changes and think that the manuscript has improved a lot. However, a fewcomments/ remarks are left: 

  1. Thank you for clarifying the study methods. As not all patients received PCR testing maybe asymptomatic COVID positive patients were categorized as false negative. Thisshould be discussed in thestudylimitations.

Thank you very much. You are certainly right, and according to your suggestion we have included in the limitations this paragraph.

“Maybe the incidence of COVID-19 among these patients was even higher since not all received PCR testing. Asymptomatic COVID positive patients could have been categorized as false negative”.

  1. Table 2: Thank you for adding the legend. The numbers seem to be moved and not at the same level, maybe the authors can improve the layout.

Thank you very much for your perception. We have corrected the edition errors, and improved the layout.

  1. Line 142: Delete „now“

Thank you very much, it has been deleted

  1. Lines 148-150: This is rather the text for the legends of the tables 1 and 2 than main-text. I would recommend to put these sentences in the legends and write the main results of table 1 and 2 in the text with referring to the tables.

Thanks once more, we have included lines of table 1 and 2 in the legends of the corresponding figures

  1. Line 160 and lines 196 ff.: As mentioned above the authors might have analyzed maybe false negative results as these patients might have had COVID but were asymptomatic. Therefore, the prevalence of COVID positive HHT patients might be even higher. It would be helpful to write the number of COVID- tests that were done and how many were positive.

Thank you. We would like to clarify that only in the 25 patients reported as COVID-19 positives we know PCR tests were done. If they were subjected to these tests was either, because they have some compatible clinical symptom, o because they belonged to a family, or group of friends were positive cases have been detected. In many other cases of the rest of 138 patients, PCRs may have been done but obviously with negative result, otherwise, they would have been included in the positive table. And viceversa, people without symptoms and without any reason to test a PCR might have suffered from asymptomatic COVID.

  1. Line 196: Here, the authors write about the incidence. Before and after they state the prevalence of COVID and HHT. Please clarify which is the correct term for the analysis the authors did.

We are grateful for this precision. We consider the COVID infection data as incidence, and HHT as prevalence. We corrected the text to be consistent.

  1. Line 257: Please change to “seemed”

Thank you, we did that

  1. Line 281: Please write “were analyzed”

Thank you, we have written the sentence in the past.

  1. Lines 291 ff: It is an interesting part of the discussion but the authors only write a general statement on treatments options for COVID-infections. I would recommend to either discuss something also related to HHT and the presented data or at least write that based on the data of this study no further analysis regarding special treatment options in HHT could be made.

The point is well taken, we decided to leave in the discussion the treatments as an interesting and general knowledge about COVID. However, and following your suggestion we have added:

“Based on the data of this study, no further analysis regarding special treatment options in HHT could be made”.

  1. Line 305: Please add “Nowadays, it is well known..”

This has been done

  1. Line 306: Please correct “common” and delete “were the most common comorbidities”

Thank you we have corrected the sentence of the line

  1. Lines 307 ff.: Please revise this very long sentence. Maybe put the information into more but shorter sentences and maybe correct the placement you used the comma.

We are grateful by your suggestion, we have revised the sentence starting in line 307 and cut the sentence putting a point in the middle of the sentence.

“Since in many cases drugs decreasing cytokines, as corticoids and tocilizumab had been used to improve the COVID-19 condition, in a way, HHT patients would be naturally producing less cytokines. Thus, without need of these treatments upon SARS-CoV-2 infection, HHT condition would avoid or smooth the acute phase, explaining the milder infections suffered by them”.

  1. Line 356: Delete „that“

Thank you very much, we have deleted “that”
